# Peroxisomal dysfunctions cause lysosomal storage and axonal Kv1 channel redistribution in peripheral neuropathy

Sandra Kleinecke[1], Sarah Richert[1], Livia de Hoz[1], Britta Brügger[2], Theresa Kungl[1], Ebrahim Asadollahi[1], Susanne Quintes[1], Judith Blanz[3], Rhona McGonigal[4], Kobra Naseri[5], Michael W Sereda[1], Timo Sachsenheimer[2], Christian Lüchtenborg[2], Wiebke Möbius[1], Hugh Willison[4], Myriam Baes[6], Klaus-Armin Nave[1], Celia Michèle Kassmann[1]*

[1]Department of Neurogenetics, Max Planck Institute of Experimental Medicine, Göttingen, Germany; [2]University of Heidelberg, Biochemistry Center (BZH), Heidelberg, Germany; [3]Unit of Molecular Cell Biology and Transgenic, Institute of Biochemistry, University of Kiel, Kiel, Germany; [4]Institute of Infection, Immunity, and Inflammation, University of Glasgow, Glasgow, United Kingdom; [5]Birjand University of Medical Sciences, Birjand, Iran; [6]Department of Pharmaceutical and Pharmacological Sciences, Cell Metabolism, KU Leuven- University of Leuven, Leuven, Belgium

*For correspondence: kassmann@ em.mpg.de

**Abstract** Impairment of peripheral nerve function is frequent in neurometabolic diseases, but mechanistically not well understood. Here, we report a novel disease mechanism and the finding that glial lipid metabolism is critical for axon function, independent of myelin itself. Surprisingly, nerves of Schwann cell-specific *Pex5* mutant mice were unaltered regarding axon numbers, axonal calibers, and myelin sheath thickness by electron microscopy. In search for a molecular mechanism, we revealed enhanced abundance and internodal expression of axonal membrane proteins normally restricted to juxtaparanodal lipid-rafts. Gangliosides were altered and enriched within an expanded lysosomal compartment of paranodal loops. We revealed the same pathological features in a mouse model of human Adrenomyeloneuropathy, preceding disease-onset by one year. Thus, peroxisomal dysfunction causes secondary failure of local lysosomes, thereby impairing the turnover of gangliosides in myelin. This reveals a new aspect of axon-glia interactions, with Schwann cell lipid metabolism regulating the anchorage of juxtaparanodal $K_v1$-channels.

## Introduction

Schwann cells, the myelinating cells of the peripheral nervous system (PNS), contain peroxisomes in distant cytoplasmic compartments, including myelin channels and cytosolic loop regions close to nodes of Ranvier (*Kassmann et al., 2011*). Peroxisomes degrade fatty acids derived from myelin lipids and generate precursors of myelin plasmalogens (*Waterham et al., 2016*). Lysosomal compartments are also present in these nodal regions (*Gatzinsky et al., 1997*), and growing evidence from in vitro studies suggests interactions between both types of organelles (*Baarine et al., 2012*; *Chu et al., 2015*; *Thai et al., 2001*).

Peroxisomal dysfunction is caused by mutations of genes encoding peroxisomal proteins or peroxisomal biogenesis factors (*Waterham et al., 2016*). In humans, *loss-of-function* mutations of ABCD1 are responsible for the disease X-linked Adrenoleukodystrophy (X-ALD). This peroxisomal ATP-binding cassette (ABC-) transporter mediates the import of very long-chain fatty acids (VLCFA)

**eLife digest** Nerve cells transmit messages along their length in the form of electrical signals. Much like an electrical wire, the nerve fiber or axon is coated by a multiple-layered insulation, called the myelin sheath. However, unlike electrical insulation, the myelin sheath is regularly interrupted to expose short regions of the underlying nerve. These exposed regions and the adjacent regions underneath the myelin contain ion channels that help to propagate electrical signals along the axon.

Peroxisomes are compartments in animal cells that process fats. Genetic mutations that prevent peroxisomes from working properly can lead to diseases where the nerves cannot transmit signals correctly. This is thought to be because the nerves lose their myelin sheath, which largely consists of fatty molecules.

The nerves outside of the brain and spinal cord are known as peripheral nerves. Kleinecke et al. have now analyzed peripheral nerves from mice that had one of three different genetic mutations, preventing their peroxisomes from working correctly. Even in cases where the mutation severely impaired nerve signaling, the peripheral nerves retained their myelin sheath.

The peroxisome mutations did affect a particular type of potassium ion channel and the anchor proteins that hold these channels in place. The role of these potassium ion channels is not fully known, but normally they are only found close to regions of the axon that are not coated by myelin. However, the peroxisome mutations meant that the channels and their protein anchors were now also located along the myelinated segments of the nerve's axons. This redistribution of the potassium ion channels likely contributes to the peripheral nerves being unable to signal properly.

In addition, Kleinecke et al. found that disrupting the peroxisomes also affected another cell compartment, called the lysosome, in the nerve cells that insulate axons with myelin sheaths. Lysosomes help to break down unwanted fat molecules. Mutant mice had more lysosomes than normal, but these lysosomes did not work efficiently. This caused the nerve cells to store more of certain types of molecules, including molecules called glycolipids that stabilize protein anchors, which hold the potassium channels in place. A likely result is that protein anchors that would normally be degraded are not, leading to the potassium channels appearing inappropriately throughout the nerve.

Future work is now needed to investigate whether peroxisomal diseases cause similar changes in the brain. The results presented by Kleinecke et al. also suggest that targeting the lysosomes or the potassium channels could present new ways to treat disorders of the peroxisomes.

into the organelle. In consequence, ABCD1-deficient peroxisomes are not capable of importing and degrading VLCFA that are specific substrates of peroxisomal $\beta$-oxidation (*Kemp et al., 2012*). A more severe impairment of peroxisomes is caused by lack of the *Hsd17b4* gene (also called multi-functional protein 2; *Mfp2* gene) that encodes a central enzyme of peroxisomal $\beta$-oxidation. In MFP2-deficient cells, the $\beta$-oxidation of virtually all peroxisome-specific substrates, including VLCFA, is inhibited (*Verheijden et al., 2013*). A complete disruption of the organelle is observed in the absence of peroxisome biogenesis factor peroxin 5 (PEX5). This cycling receptor recognizes proteins with a peroxisomal targeting sequence type 1 (PTS1) and is involved in their transfer into peroxisomes. PEX5-dependent protein import applies to the majority of peroxisomal enzymes. Thus, PEX5-deletion disrupts peroxisomal function substantially (*Waterham et al., 2016*).

Schwann cell lipid metabolism is rate-limiting for myelination and is important for maintenance of axonal integrity (*Saher et al., 2011*; *Viader et al., 2013*), which requires in addition to membrane wrapping the assembly of nodal, paranodal, and juxtaparanodal membrane proteins (*Rasband and Peles, 2015*). The juxtaparanodal domain of myelinated axons harbors voltage-gated *shaker-type* potassium channels, $K_v1.1$ (KCNA1) and $K_v1.2$ (KCNA2; *Chiu and Ritchie, 1980*; *Robbins and Tempel, 2012*), which also align the inner mesaxon as a thin band (*Arroyo et al., 1999*). Associated with connexin-29 hemichannels (*Rash et al., 2016*), their clustering and anchoring at juxtaparanodes requires the neuronal membrane proteins CASPR2 and TAG-1, the latter expressed by glia and neurons (*Poliak et al., 1999b*; *Traka et al., 2003*). $K_v1$ channels have been proposed to play a role in regulating fiber excitability (*Baker et al., 2011*; *Glasscock et al., 2012*), but the exact in vivo

function of these fast-opening/slowly inactivating channels remains unknown (*Arancibia-Carcamo and Attwell, 2014*).

## Results

*Cnp-Cre::Pex5$^{flox/flox}$* mice, termed cKO or 'mutants' in the following, lack peroxisomal protein import in Schwann cells (*Figure 1a*; *Figure 1—figure supplement 1a*). The PNS of these mice is well myelinated and unlike the CNS (*Kassmann et al., 2007*) without immune-mediated injury, in agreement with pilot observations (*Kassmann et al., 2011*). Upon closer inspection, we determined about 50% genomic recombination, corresponding to the fraction of Schwann cell (SC) nuclei in sciatic nerves (*Figure 1—figure supplement 1b*). Teased fiber preparations, stained for PMP70, revealed peroxisomes as puncta. In mutant nerves, these were import-deficient 'ghosts', as evidenced by cytoplasmic catalase, normally a luminal peroxisomal marker (*Figure 1b*).

Peroxisomal dysfunction in myelinating SC was confirmed by lipid mass spectrometry (*Figure 1c*, *Figure 1—figure supplement 1c*), showing reduced plasmalogens (PE$_P$-) and its precursor alkylated phosphatidyl-ethanolamines (PE$_{O-}$; *Wanders, 2014*). Also VLCFA were increased, indicating the accumulation of peroxisomal $\beta$-oxidation substrates (*Figure 1d*; *Figure 1—figure supplement 1d*).

We determined nerve conduction velocity (NCV) by electrophysiology of isolated sciatic nerves (to avoid possible contributions of altered muscle endplates) at the age of 2 months (*Figure 1e–g*; *Figure 1—figure supplement 2a*). For all stimulus intensities tested, responses of mutant nerves were different from controls (*Figure 1—figure supplement 2b*). Compound action potentials (CAPs) and NCV were diminished in mutants (mean: 28 ± 4.7 m/s) compared to controls (41.5 ± 3.6 m/s; *Figure 1e*). Thresholds to evoke a signal were only slightly elevated (155μA versus 135μA), but the maximal response was 50% of control (*Figure 1f,g*). Also, in vivo recordings revealed significantly reduced compound muscle action potentials (CMAPs) in mutant mice (*Kassmann et al., 2011*). This was more enhanced when stimulating proximally than distally, which clinically defines conduction blocks (*Figure 1—figure supplement 2c*).

We suspected that reduced nerve conduction would be explained by demyelination. Surprisingly, by immunohistochemistry and Western blot analysis structural myelin proteins, including PMP22, MPZ/P0, and P2, were not significantly altered (*Figure 1—figure supplement 3a,b*). Only PLP, a minor PNS myelin protein, showed significant reduction. Also by electron microscopy (EM), myelin thickness, periodicity, and compaction were indistinguishable (*Figure 1h*, *Figure 1—figure supplement 3c*). Next, we determined internodal length in teased fiber preparations, which was shorter in mutant (623 nm) than in control fibers (691 nm; *Figure 1—figure supplement 3d*), but unlikely sufficiently reduced to cause a slower conduction by itself (*Wu et al., 2012*). Importantly, while the reduced CAP suggested significant axon loss at 2 months, the morphometric analysis of entire nerve cross-sections revealed normal axonal numbers and calibers (*Figure 1i*). Thus, reduced CAPs were most likely caused by functional conduction blocks rather than physical axon loss. Signs of axonal transport defects (APP accumulations) were not observed (*Figure 1—figure supplement 3e*), and also, indirect signs of neurodegeneration (neuroinflammation) were not significantly enhanced in nerves at 2 months (*Figure 1—figure supplement 4a–d*).

Disruptions of axo-glial junctions might cause leak currents and thus conduction failure (*Arancibia-Carcamo and Attwell, 2014*). Yet, both neurofascin-155 (NF155) and axonal CASPR, that is adhesion proteins that mediate axo-glial contacts (*Desmazieres et al., 2014*; *Normand and Rasband, 2015*), were normally localized (*Figure 1j*). By EM, we never found the detachment of paranodal loops (*Figure 1k,l*), even where abnormally enlarged by accumulated vesicles (*Kassmann et al., 2011*). Next, we examined the distribution of ion channels (*Devaux et al., 2004*; *Rasband and Peles, 2015*) in sciatic nerve teased fibers from 2- and 9-month-old mutants and controls. Nodal Na$_v$1.6 and K$_v$7.2, as well as their anchoring proteins, were normally flanked by CASPR-positive paranodes (*Figure 2—figure supplement 1a–c*). Surprisingly, the juxtaparanodal potassium channel protein K$_v$1.1 displayed an abnormal localization, frequently displaced into the adjacent internodal region, and to a minor extent into paranodes. K$_v$1.1 clusters were in the majority of cases still maintained at juxtaparanodes, suggesting that internodal mislocalization is secondary (*Figure 2—figure supplement 1d,e*; *Figure 2a*). Indeed, the pathology of internodal K$_v$1 channels progressively increased with age frequently presenting more than one extra cluster along one internode in aged mutants, while shifts into the nodal direction were not progressive (*Figure 2a,b*; *Figure 2—figure*

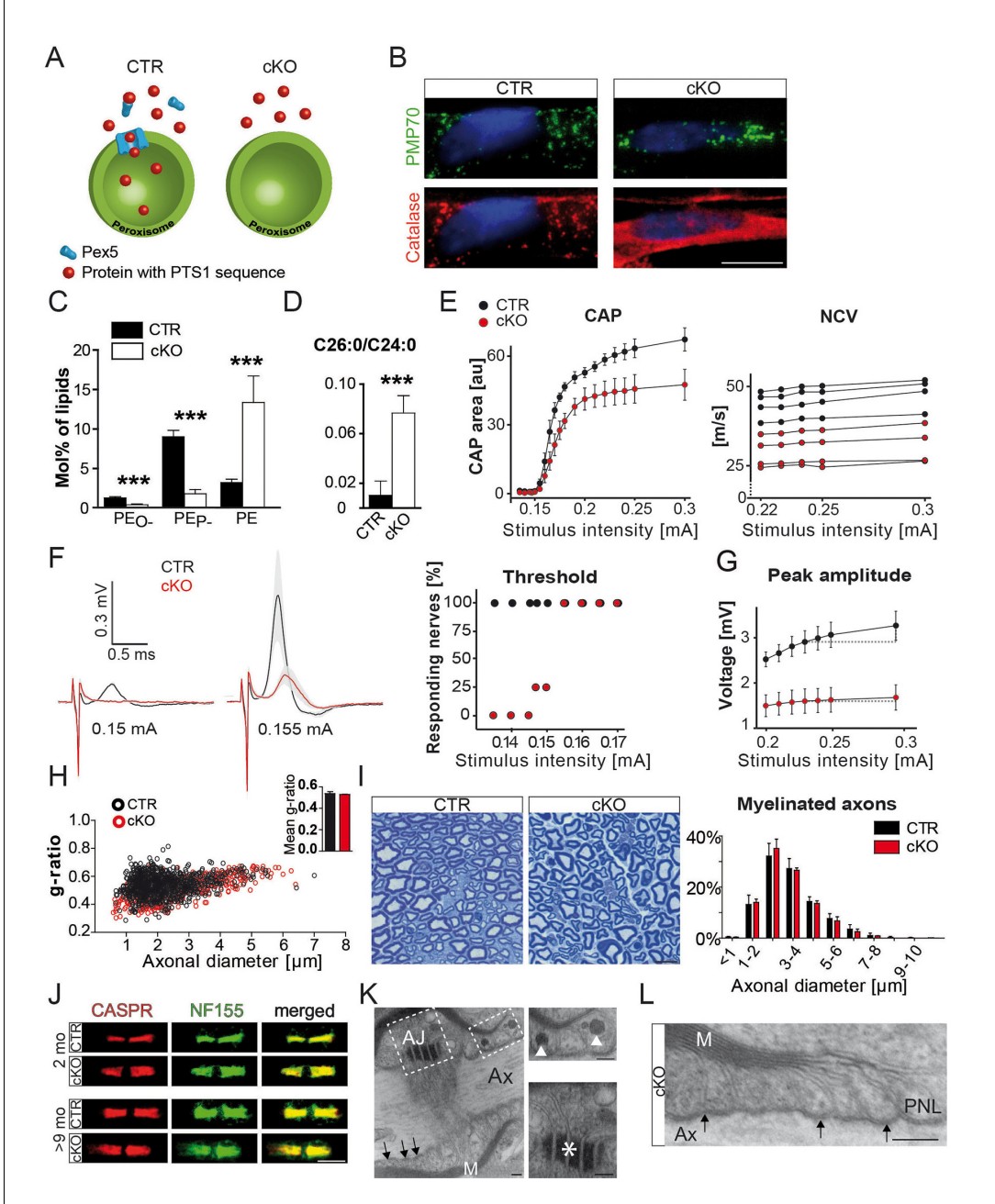

**Figure 1.** Schwann cell-specific PEX5-deficiency causes peroxisome dysfunction and peripheral neuropathy in the absence of axonal loss or dysmyelination. (A) Scheme of normal (left) and impaired (right) PEX5-dependent peroxisomal protein import. PTS1, peroxisomal targeting signal type 1; PEX5, peroxisomal biogenesis factor peroxin 5. (B) Catalase (red) is present in peroxisomes (PMP70; green) of controls, but is localized in the cytoplasm of mutant fibers. PMP70, peroxisomal membrane protein 70; DAPI-stained SC nuclei are depicted in blue; Scale bar, 10 μm. (C, D) Lipid mass spectrometry of nerve lysates from controls and mutants aged 9 months indicates peroxisomal dysfunction. Peroxisomal products (PE$_{O-}$) and its corresponding plasmalogens (PE$_{P-}$) are reduced. Specific substrates of peroxisomal $\beta$-oxidation, VLCFA, are accumulated in mutant nerves. Statistics: means ± s.d.; n = 6; Student's T-test; ***p<0.001. (E–G) Functional impairment of mutant compared to control sciatic nerves is assessed by electrophysiology at the age of 2 months (n = 4). To evoke significant responses of all measured nerves, larger stimulus intensity is required for mutant (0.155mA) as compared to control (0.135mA) nerves. Peak amplitudes plotted against increasing stimulus intensity indicates earlier response saturation of mutant nerves. CAP, compound action potentials; au, arbitrary units; NCV, nerve conduction velocities (each curve representing single-nerve mean responses across intensities). (H) g-ratio analysis by electron microscopy of sciatic nerves as measure of myelin thickness (n = 3; ≥100 randomly chosen axons per nerve at age 2 months). (I) Axonal analysis by methylene blue-stained semithin cross-sections of sciatic nerves of control and mutant mice at the age of 2 months (n = 3). Scale bar, 10 μm. (J) Immunostained teased sciatic nerve fibers of mutants and controls at 2 and 9 months show normal paranodal localization of CASPR (red) and anchoring protein neurofascin (NF155; green). (K, L) Intact transverse bands (arrows) at paranodal loops
*Figure 1 continued on next page*

*Figure 1 continued*

(PNLs) between axon (Ax) and myelin (M) by electron microscopy in a 9-month mutant nerve, even when PNLs harbor enlarged vesicles (top inset, arrowheads). Mutant nerves display normal adherens junctions (AJ) between loops (bottom inset, asterisk). Scale bars, 500 nm; scale bars in insets, 250 nm.

The following figure supplements are available for figure 1:

**Figure supplement 1.** Successful PEX5 ablation causes peroxisomal dysfunction.

**Figure supplement 2.** Electrophysiology of mouse sciatic nerves.

**Figure supplement 3.** Myelin analysis of mouse sciatic nerves.

**Figure supplement 4.** Young *Pex5* mutant nerves lack signs of inflammation.

*supplement 1e*). We confirmed the observation of displaced juxtaparanodal $K_v1$ channels using $K_v1.2$ antibody (data not shown). At 9 months, we determined a twofold overall increase of $K_v1.1$ in nerve lysates (*Figure 2c,d*). However, as most $K_v1.1$ is expressed by unmyelinated fibers, the increase of internodal $K_v1.1$ in myelinated axons may be significantly higher. Neuronal CASPR2, the *cis*-anchor for $K_v1$ channels in the axonal membrane, and TAG-1 on SC are essential for channel clustering at juxtaparanodes (*Hivert et al., 2016*; *Poliak et al., 1999a*; *Savvaki et al., 2010*). Both proteins colocalized with $K_v1.1$ at regular and ectopic internodal positions (*Figure 2e–g*). TAG-1 expression was slightly but not significantly elevated (*Figure 2h*). In contrast, staining for myelin-associated glycoprotein (MAG) did not reveal a connection between Schmidt-Lanterman incisures and the ectopic $K_v1.1$/TAG-1/CASPR2-positive clusters (data not shown). Although, we cannot exclude that enhanced expression of TAG-1 contributes to the formation of ectopic protein clusters, the observations suggest that in mutant nerves entire membrane patches of the juxtaparanodal compartment accumulate, break-off, and drift into the internodal region.

Gangliosides are glycosphingolipids important for stabilizing membrane-spanning proteins and axon-glia interactions (*Susuki et al., 2007*). We analyzed sciatic nerve lysates from 9-month-old mutants by lipid mass spectrometry and noted that many more gangliosides contained VLCFA (a 14-fold enrichment; *Figure 2h*). To determine their subcellular distribution, we stained teased fibers with a fluorescently tagged cholera-toxin subunit B (CTB), which binds GM1 gangliosides. Also by immunostaining, GM1 was restricted to paranodes in wild-type nerves but widely dispersed into the internodal myelin of mutant nerves (*Figure 2—figure supplement 2*; *Figure 2i*). Likewise, ganglioside GD1a could be immunostained as enlarged puncta (>5 μm) in internodal myelin of mutant nerves (*Figure 3a*). These GD1a-containing vesicles that colocalized with LAMP1 were a frequent finding in mutant internodes, but rarely observed in controls (*Figure 3a*, left).

Paranodal loops in conditional mutants showed vesicular inclusions of variable size and electron-density (*Figure 3b*). By immune-labeling of ultra-thin cryosections and teased fibers, the majority of these were LAMP1-positive (*Figure 3c,d*; *Figure 3—source data 1*). Even more striking, lysosomal LIMP-2 marked small puncta in controls (<500 nm), but giant compartments within mutant paranodes (*Figure 3e*, right). Strongly increased abundance of LAMP1 and LIMP-2 was confirmed by Western blotting (*Figure 3f,g*). Decreased mRNA levels (*Figure 3h*) suggest that the enrichment of LAMP1 and LIMP-2 marks organelle accumulation rather than enhanced lysosomal biogenesis.

Lysosomal markers are found in autophagosomes and early/late endosomes (*Saftig and Klumperman, 2009*). We thus phenotyped accumulated vesicles with antibodies specific for EEA1, RAB7, and ATG5. Neither marker was increased at mutant paranodes (*Figure 3—figure supplement 1a–c*). Therefore, the majority of accumulating vesicles within mutant paranodes are likely *bona fide* lysosomes, serving the degradation of myelin-associated proteins/lipids, including gangliosides (*Luzio et al., 2007*; *Saftig and Klumperman, 2009*). To determine the relationship between paranodal lysosomes (*Gatzinsky et al., 1997*) and peroxisomes, we immunostained for PMP70 and LAMP1, revealing a close spatial association (*Figure 3i*) as reported for cultured cells (*Chu et al., 2015*). We determined higher activities of lysosomal α-mannosidase and β-hexosaminidase in mutant nerve lysates (*Figure 3j*), and note that increased compartment size and enzymatic activities are known

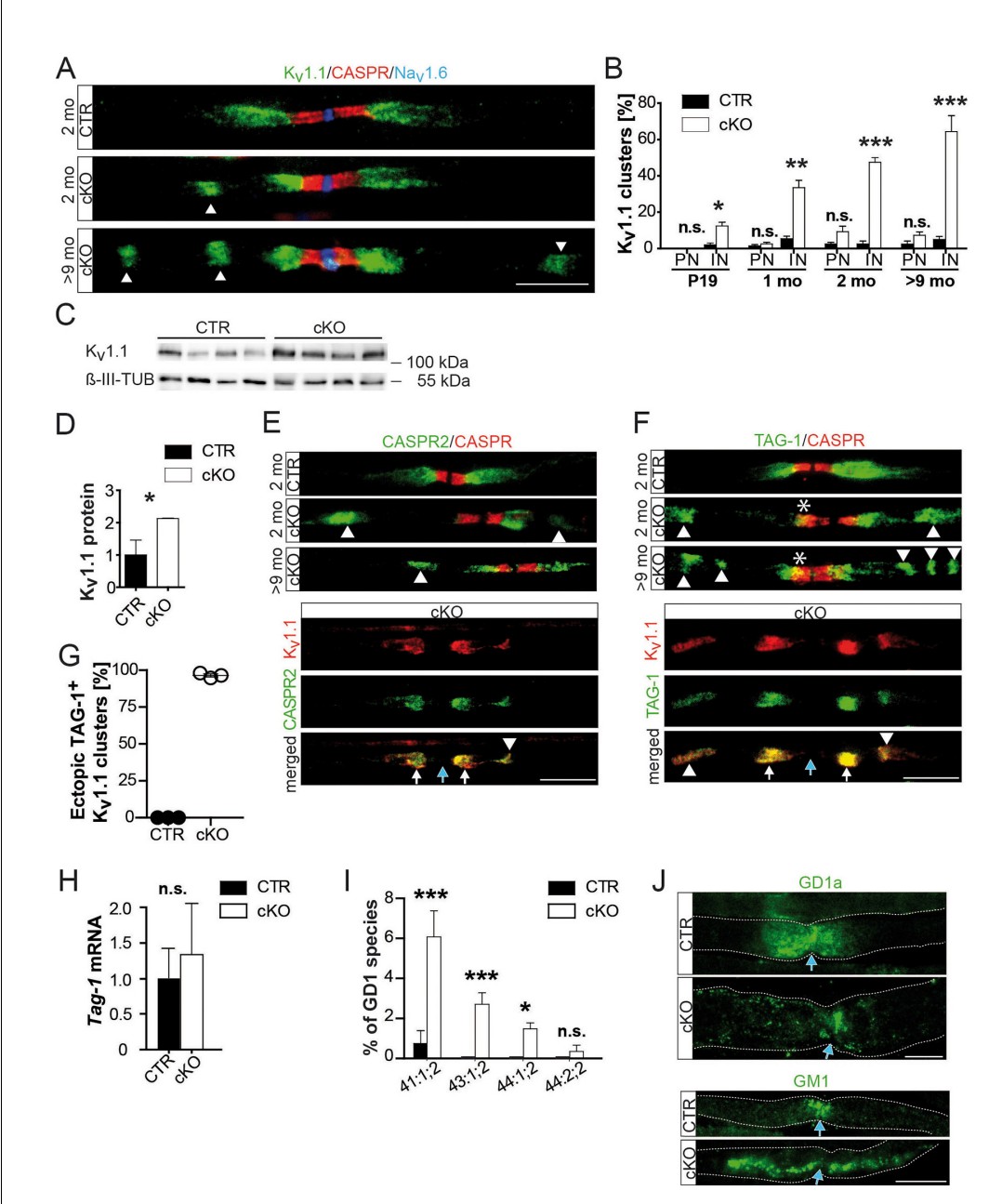

**Figure 2.** Abnormal distribution of axonal ion channels $K_v1.1$, its anchoring proteins, and associated lipids in conditional *Pex5* mutant nerves. (**A, B**) Immune-stained sciatic teased fibers of 2- and 9-month-old mice reveal normally localized nodal sodium channels ($Na_v1.6$; blue) flanked by paranodal CASPR (red; scale bar, 10 µm). Only mutant nerves show extra $K_v1.1+$ patches within internodes. This is progressive with age shown by the corresponding quantification at different ages, starting at P19, revealing a shift of $K_v1.1$ primarily to internodes (IN), and (not significantly) to paranodes (PN). n = 3 for P19 and 1 month; n = 5 for 2 months; n = 7 for>9 months. Statistics: means ± s.e.m, two-way ANOVA followed by Student's T-test. *p<0.05; **p<0.01; ***p<0.001; n.s., not significant. (**C, D**) Western blot analysis of sciatic nerve lysates of 9-month-old mice shows increased $K_v1.1$ protein abundance in mutant nerve lysates. *β*-III Tubulin served as loading control for normalization (n = 4). (**E–G**) $K_v1$-anchoring proteins (green) CASPR2 and TAG-1 showed in addition to normal juxtaparanodal localization an ectopic internodal (arrow heads) and occasionally a partial overlap (asterisk) with paranodal CASPR (red) in mutant sciatic teased fibers. Both proteins colocalized at original (white arrows) and ectopic (arrow heads) positions (bottom panels). Quantification of TAG-1/$K_v1.1$-positive ectopic clusters is depicted, (**G**). Blue arrows indicate nodes of Ranvier. Scale bars, 10 µm. (**H**) qRT-PCR analysis (n = 5) for TAG-1 showing a tendency of increased mRNA expression in mutant compared to control nerves at 2 months. (**I**) Mass spectrometry of sciatic nerve lysates (n = 3) obtained at the age of 9 months shows percentage of GD1 species per genotype containing 41, 43, or 44 C-atoms (':' separates number of double-bonds and ';' separates number of hydroxyl-groups). Only 0.74% of GD1 in control lysates contain 41 carbons, species with more carbons are absent, while highly enriched in mutant nerve lysates. (**J**) Antibody-staining for gangliosides of nerves from mice

*Figure 2 continued on next page*

*Figure 2 continued*

aged 9 months shows GD1a+ vesicles abnormally distended into mutant internodes (green, top). GM1 is localized at paranodes of control fibers, but extends into mu internodes. Dotted lines indicate borders of myelinated fibers. Blue arrows indicate nodes of Ranvier. Scale bars, 10 µm. Statistics: means ± s.d.; *p<0.05; **p<0.01; ***p<0.001; n.s., not significant, Student's T-test.

The following figure supplements are available for figure 2:

**Figure supplement 1.** Normal distribution of nodal proteins and associated anchors in *Pex5* mutant nerves.

**Figure supplement 2.** Abnormal distribution of GM1 ganglioside in the perinodal region of mutant nerves by CTB-staining.

phenomena of lysosomal disorders (*Sardiello and Ballabio, 2009*), yet associated with organelle dysfunction.

Theoretically, complete peroxisomal failure in *Cnp-Cre::Pex5$^{flox/flox}$* mice could perturb many aspects of SC metabolism in the observed neuropathy. However, we found that *Cnp-Cre:: Hsd17b4$^{flox/flox}$* mice (referred to as MFP2 conditional knockout mice in the following), which specifically lack peroxisomal $\beta$-oxidation (*Verheijden et al., 2013*), share the key features of this novel phenotype, including ectopic internodal $K_v1.1$ clusters (*Figure 4—figure supplement 1a–e*), confirming the important role for glial lipid metabolism in maintaining axonal membrane composition, and function.

To investigate whether this new pathomechanism, identified in two mouse models with a glia-specific mutation, is also relevant for a human genetic disease, we analyzed ABCD1-deficient mice, a model for X-ALD/AMN (*Forss-Petter et al., 1997*). Sciatic nerve axons and myelin were morphologically intact, even at 22 months of age (*Figure 4a–c*). However, ABCD1-mutants exhibited a similar mislocalization of $K_v1.1$ positive channels as observed in both conditional (*Pex5* and *Mfp2*) mutants. The quantification showed that mislocalization of internodal $K_v1.1$ was temporally progressive. In accordance with our model of membrane patches drifting into internodes, significant alterations were observed only at 22 months of age, but not yet at 2 months (*Figure 4d–e*). This pathology was accompanied by lysosomal accumulates in myelinated fibers (*Figure 4f–h*).

## Discussion

Our data suggest an important pathomechanism of peroxisomal dysfunction in peripheral nerves is the secondary impairment of lysosomes. For example, gangliosides, which are frequently esterified with VLCFA and 2-hydroxy fatty acids (*Chrast et al., 2011*; *Sandhir et al., 2000*), must be degraded in lysosomes. However, subsequent $\beta$-oxidation of VLCFA and 2-hydroxy fatty acids requires peroxisomes (*Foulon et al., 2005*; *Wanders, 2014*). Thus, our mouse model provides first in vivo evidence for functional interactions between lysosomes and peroxisomes, so far only discussed for cultured cells (*Chu et al., 2015*).

One consequence of lysosomal impairment is the perturbation of normal ganglioside turnover. Gangliosides are in close association with $K_v1$-anchoring proteins (*Loberto et al., 2003*) and provide stability to the juxtaparanodal clusters (*Susuki et al., 2007*), which resemble giant membrane 'lipid rafts'. As mechanism for ectopic localization of axonal $K_v1$ channels, we suggest the following working model (*Figure 4—figure supplement 2*): Lack of ganglioside breakdown causes their accumulation in the glial juxtaparanodal membrane. Thereby, clusters of $K_v1$ and associated anchoring proteins (TAG-1 and CASPR2) on the axonal side gain abnormal stability, break-off, diffuse into the internodal region, and escape from regular turnover. It is likely that also glial TAG-1 remains anchored to the axonal $K_v1$-/CASPR2-/TAG-1 clusters, but this cannot be distinguished by TAG-1 immunofluorescence.

$K_v1$-channels are fast-opening, voltage-gated channels that are activated by mild voltage-shifts and thought to regulate fiber excitability (*Glasscock et al., 2012*). The integrity of $K_v1$ channels is also impaired in CASPR2-/- mice, but perturbation of nerve function is not a feature (*Poliak et al., 2003*). Unlike PEX5 mutant nerves, CASPR2-/- nerves do not exhibit elevated levels of $K_v1.1$ channel protein. Also, the ectopic $K_v1.1+$ clusters are situated directly adjacent to juxtaparanodes in CASPR2-deficient fibers, not along the internodes as observed in PEX5 mutants. Considering the

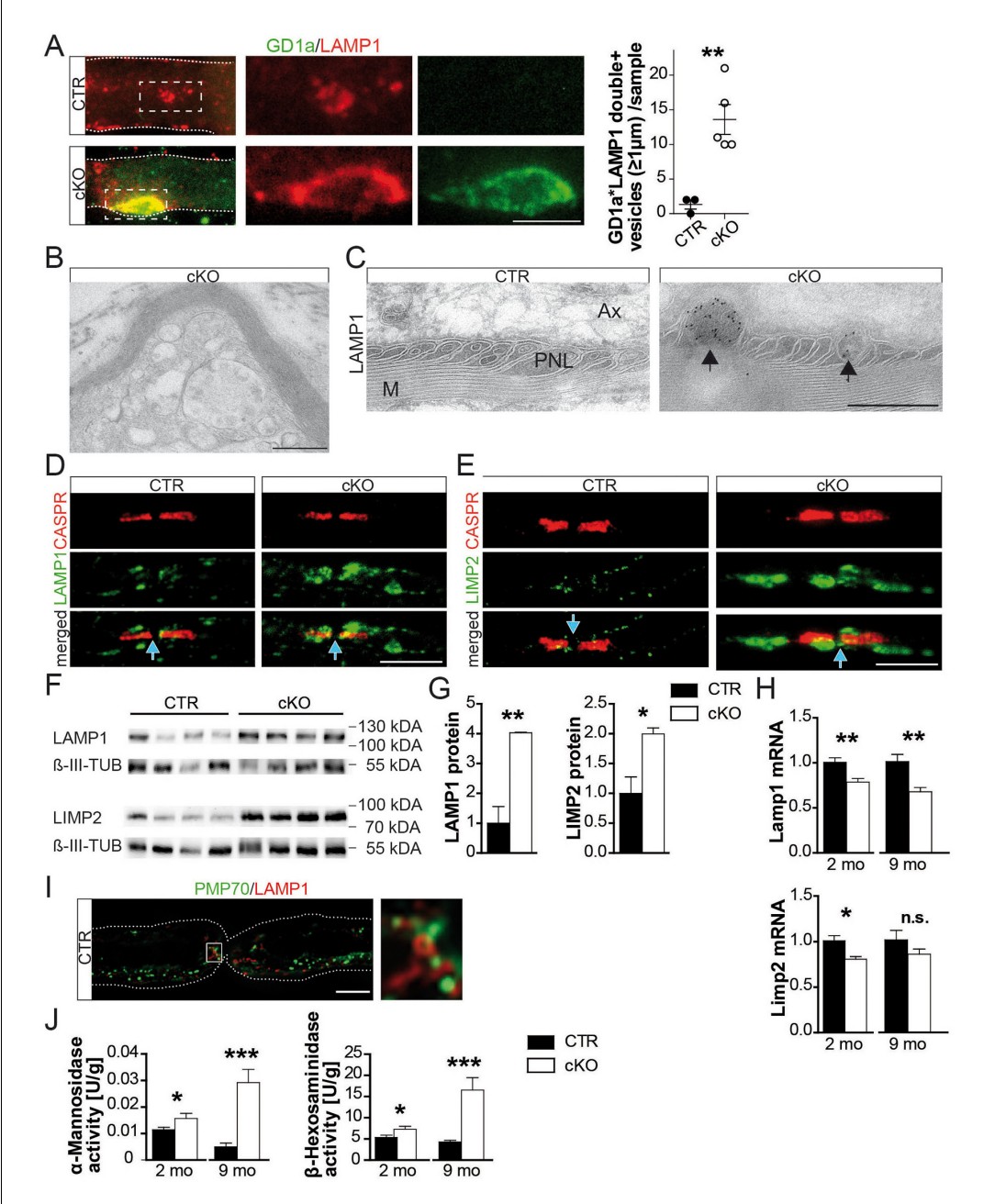

**Figure 3.** Hallmarks of lysosomal storage disorders in *Pex5* mutant nerves. (A) Double-staining of sciatic teased fibers displayed giant GD1a+ (green) vesicles that colocalized with LAMP1 (red) in internodes only of mutant animals (rectangular inset magnified in right panels). Dotted lines indicate borders of myelinated fibers; scale bar in inset, 5 μm. (B) Electron microscopy of a teased and sectioned mutant sciatic nerve depicting paranodal vesicle accumulations. Scale bar, 500 nm. (C) Immuno-electron microscopy of sciatic nerves from mice aged 9 months identifies enlarged LAMP1+ vesicles (arrows) within paranodal loops (PNLs). M, myelin; Ax, axon. Scale bar, 500 nm. (D, E) Immunolabeling of teased sciatic nerves at the age of 2 months shows enlarged LAMP1+ (left, green) or LIMP-2+ (right, green) compartments close to CASPR+ paranodes (red) of mutant nerves. Scale bars, 10 μm. (F–H) Analysis of sciatic nerve lysates by Western blotting (n = 4; β-III Tubulin, served as loading control for normalization) and qRT-PCR (n = 6) for LAMP1 and LIMP-2 showing increased protein, but decreased mRNA levels in mutants. (I) Immunostaining of control sciatic fibers shows close association of peroxisomal PMP70 (green) and LAMP1 on lysosomal membranes (red) in the paranodal region, suggesting physical interactions. A wide-field flourescent image after deconvolution (250 nm z-stack width) is shown. Scale bar, 10 μm. Dotted lines indicate borders of the myelinated fiber. (J) Lysosomal enzyme activities of nerve lysates was enhanced in mutants compared to controls as assessed by assays using substrates of α-Mannosidase (left) and β-Hexosaminidase (right; n = 4). Statistics: means ± s.e.m. *p<0.05; **p<0.01; ***p<0.001; n.s., not significant, Student's T-test (G, H, J). Blue arrows indicate nodes of Ranvier (D, E, I).

*Figure 3 continued on next page*

*Figure 3 continued*

The following source data and figure supplement are available for figure 3:

**Source data 1.** Lysosomal accumulation within paranodal loops.

**Figure supplement 1.** Characterization of paranodal vesicle inclusions in *Pex5* mutant nerves.

severe progressive decline of nerve function in PEX5 mutant mice, which becomes clearly more affected with age (data not shown), it is unlikely that a rather non-progressive pathology, that is the shift of $K_v1$ into paranodes, is the underlying cause of conduction slowing. In contrast, the progressive shift of $K_v1$ toward internodes correlates well with disease progression. Theoretically, $K_v1$-mediated currents in the internode could hyperpolarize the axonal membrane and dampen the conductivity of spiking axons. Such 'feedback regulation' might prevent hyperexcitability, which is a feature of mice lacking $K_v1.1$ function (*Zhou et al., 1998*). On the other hand, a 'gain-of-function' effect by opening additional ectopic internodal $K_v1$ channels, may contribute to a disturbed equilibrium of axonal ions causing slowed NCV and conduction blocks. Direct proof of this model would require patch clamping single axons underneath myelin, which is impossible at present.

Peripheral neuropathy has been associated with numerous metabolic diseases, including AMN. In this clinically milder variant of ABCD1 loss-of-function mutations, the reduced NCV was assumed to reflect demyelination, which is a hallmark in the brain of severely affected patients. Similarly, conditional PEX5 knockout mice display a cerebral inflammatory demyelination, while such pathology is absent in sciatic nerves. This remarkable discrepancy between the peripheral and central nervous system pathology is currently unexplained and presumably related to differently responding cell populations (*Kassmann, 2014*). Also in ABCD1-deficient nerves neither demyelination nor axon degeneration, but ectopic axonal $K_v1$ channels was a progressive pathological feature as early as at 2 months of age, preceding the disease onset by more than 1 year (*Pujol et al., 2002*). Our present findings therefore expand the emerging role of myelinating glial cells in maintaining axonal membrane composition and thereby influencing axonal function, independent of myelin itself.

## Materials and methods

### Animals

*Cnp1^{Cre/+}* and *Pex5^{flox/flox}* mice with C57/Bl6 genetic background were genotyped as described (*Kassmann et al., 2007*). To check for recombination, genomic DNA was isolated from tail biopsies using the nexttec kit according to the manufacturer's instructions. PCR was performed with sense (5'- CCAACGTCTGCCCATTCCTCCACCTG-3') and antisense primers (5'- TTTGAGGATGGGAAG-CAGTGCT −3') generating a 330 bp amplicon after Cre-mediated excision of the floxed *Pex5* gene in conditional knockout mice. *Hsd17b4^{flox/flox}* (*Mfp2^{flox/flox}*) mice and *Abcd1* knockout mice with C57/Bl6 genetic background were genotyped by PCR as described (*Verheijden et al., 2013*; *Forss-Petter et al., 1997*). Animals were maintained in individually ventilated cages under SPF conditions.

### Immunofluorescence

For visualization of gangliosides sciatic nerves of *Cnp-Cre::Pex5^{flox/flox}* mice and controls were dissected and pre-teased in artificial cerebrospinal fluid (ACSF; 126 mM NaCl, 3 mM KCl, 1,25 mM NaH2PO4, 26 mM NaHCO3, 2 mM MgSO4, 10 mM Glucose, 2 mM CaCla2). Fiber bundles were incubated for 1.5 hr at room temperature with antibodies specific for GM1 or GD1a (kindly provided by Hugh Willison, Glasgow). Samples were washed in ACSF, incubated with isotype-specific fluorescent secondary antibodies (Alexa-488 IGg3 and IG2b goat anti-mouse; Thermo Fisher Scientific, Massachusetts, USA), again washed in ACSF, fixed in 4% paraformaldehyde (PFA) for 5 min, then washed in PBS containing 0.1 M glycine, teased on glass slides, air-dried, and stored at −20°C. For other antibody-labeling, sciatic nerves were dissected, teased and stored at −20°C as described previously (*Kassmann et al., 2011*). For staining protein antigens, frozen teased fibers were fixed in 4% paraformaldehyde (PFA) for 5 min, permeabilized in methanol at −20°C for 3–5

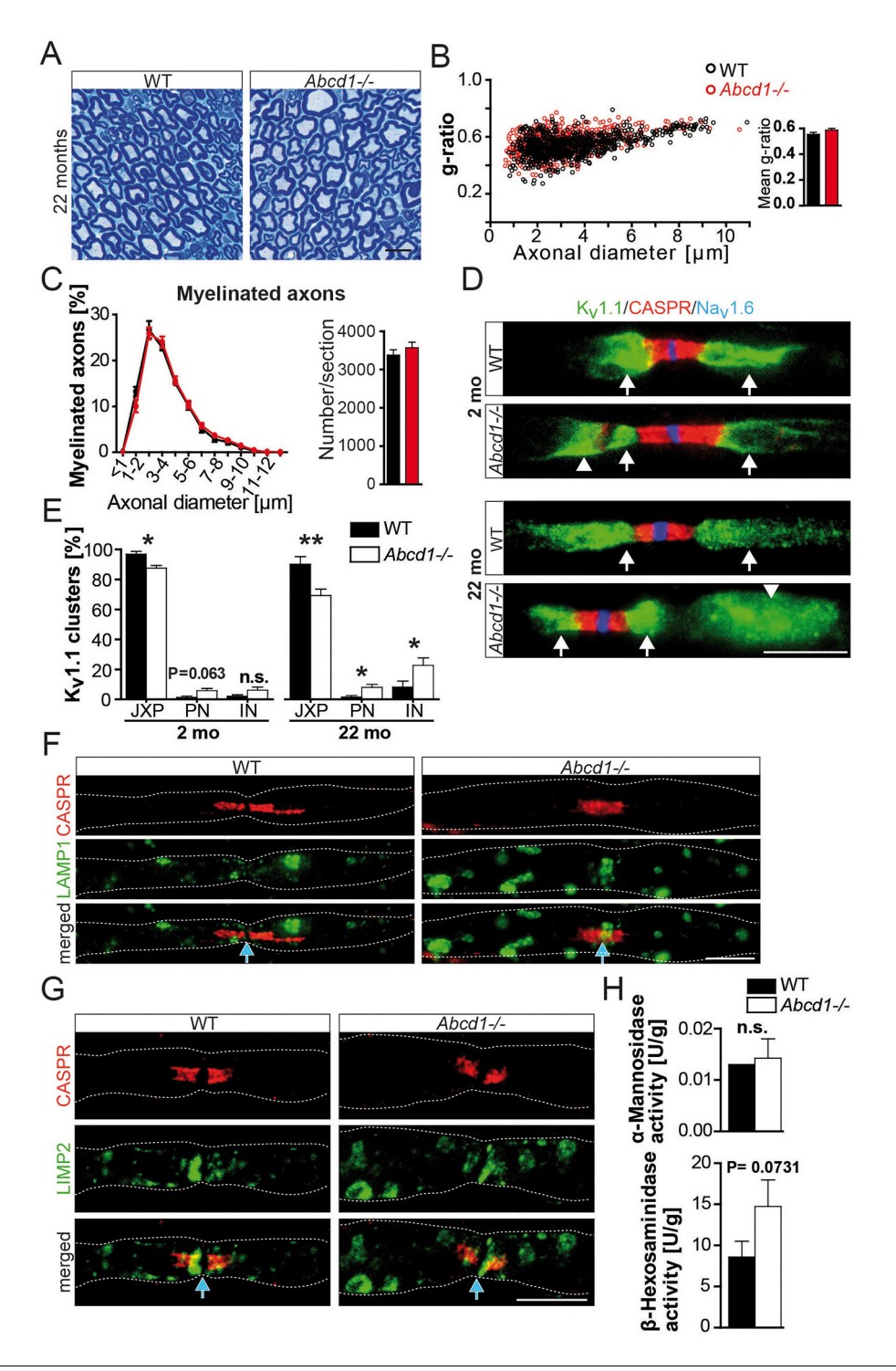

**Figure 4.** Normal nerve morphology, but ectopic ion channels and lysosomal accumulates in 22 months aged *Abcd1* null mutants. (**A–C**) Analysis of nerve morphology using entire methylene blue-stained semithin cross-sections (**A, C**) and g-ratios as measure of myelin thickness by electron microscopy of sciatic nerves (≥200 randomly chosen axons per nerve) obtained from control and *Abcd1* knockout mice (n = 4). Scale bar, 10 μm. (**D, E**) Immune-stained teased sciatic nerve fibers obtained from mutants and controls with corresponding quantifications show progressively abnormal K$_v$1.1 localization (green), while Na$_v$1.6 (blue) and CASPR (red) are normally localized. Scale bars, 10 μm. (n = 4, 2 mo; n = 6, 22 mo). Statistics: means ± s.e.m, Student's T-test *p<0.05; **p<0.01; ***p<0.001; n.s., not significant. (**F–H**) Analysis of lysosomes reveals LAMP1+ and LIMP-2+

*Figure 4 continued on next page*

*Figure 4 continued*

accumulations (green) by immune-stained sciatic teased fibers (n = 3) and a mild increase of lysosomal $\beta$-Hexosaminidase activity measured using sciatic nerve lysates (n = 4). Blue arrows indicate nodes of Ranvier. Scale bars, 10 µm. Statistics: means ± s.e.m, two-way ANOVA followed by Bonferroni (**B**) or Student's T-test (**H**). *p<0.05; **p<0.01; ***p<0.001; n.s., not significant.

The following figure supplements are available for figure 4:

**Figure supplement 1.** Ectopic, internodal $K_v1$-channel clusters in nerves of conditional MFP2 knockout mice.
**Figure supplement 2.** Hypothetical model for lipid turnover of the juxtaparanodal anchoring complex of $K_v1$-channels.

min, PBS-washed, and blocked for 1 hr in PBS containing 10% horse serum and 0.05% Triton X-100. Primary antibodies diluted in blocking solution were applied overnight at 4°C, specimens were washed in PBS, and incubated for 1 hr with fluorescent secondary antibodies Alexa-488, Alexa-555 (Thermo Fisher Scientific, 1:2000), and DyLight 633 (1:1000, YO Proteins, Sweden). Cell nuclei were visualized with 4', 6'- diamidino-2-phenylindole (DAPI; 1:10.000, Thermo Fisher Scientific). Fibers were mounted in AquaPolymount (Polysciences, Pennsylvania, USA). Fluorescent images were acquired with an inverted Zeiss Axio Observer.Z1 equipped with an Axiocam MRm (Zeiss, Germany).

## Immunohistology

Mouse sciatic nerves were fixed overnight in 4% PFA, embedded in paraffin, and longitudinally sectioned (5 µm). For labeling, Dako LSAB2 system (Dako, Denmark) was used according to the manufacturer's directions. All samples were analyzed by light microscopy (Zeiss Axiophot).

## Semithin sections and electron microscopy

Mice were anesthetized with avertin (100 µl/10 g bodyweight) and perfused intracardially with 15 ml HBSS (Lonza, Switzerland), followed by 30 ml of fixative (0.2% glutaraldehyde, 4% paraformaldehyde in PBS). Sciatic nerves were dissected and post-fixed for 2–4 hr in 4% followed by overnight incubation in 1% PFA. Epon embedding of sciatic nerves was performed as described previously (*Kassmann et al., 2007*).

Semithin cross-sections (500 nm) of sciatic nerves were incubated with methylene blue/Azur II and axons of entire cross-sections of sciatic nerves were analyzed by light microscopy (Zeiss Axiophot). Quantification of axon number was performed using Fiji software.

Preparation of ultrathin cryosections (50 nm) was performed according to the Tokuyasu technique, and immunolabeling was carried out as previously described (*Werner et al., 2007*). Antibody binding was visualized with protein A-gold (1:60, 10 nm).

Ultrathin sections (50 nm) were contrasted with 1% uranyl acetate and lead citrate. Sections were examined with a LEO EM 912AB electron microscope (Zeiss), and pictures were taken with an on-axis 2048 X 2048 CCD camera (Proscan). Myelin thickness was evaluated using Fiji software by g-ratio analysis (dividing the axonal by the fiber diameter) derived from at least 100 randomly chosen fibers per nerve.

## Protein biochemistry

Sciatic nerves were transferred to 200 µl (per nerve) ice-cold lysis buffer containing 1% Triton X-100, 50 mM Tris-HCl pH 7.5, 150 mM NaCl, 1% NP-40, and protease inhibitors (Complete, Roche, Switzerland). Samples were homogenized using Precellys 24 (VWR International, Dealware, USA), centrifuged for 10 min at 1000 g to remove cellular nuclei, and protein concentration of the supernatant was determined using Lowry assay (Bio-rad Laboratories, California, USA). Blotting membranes were incubated overnight at 4°C with primary antibodies. Western Lightning + ECL Kit (Perkin Elmer, Massachusetts, USA) was used to label protein bands, which were detected using Intas ChemoCam Image system. At least four biological replicates were analyzed. Quantification was performed in Fiji.

## Antibodies

4.1G (1:100; kindly provided by E. Peles) AnkG (1:100; kindly provided by M. Rasband), APP (1:1000; MAB348, Merck Millipore, Massachusetts, USA), ATG5 (1:100; pab50264 Covalab, France), CASPR (mk, m; 1:1000; clone K65/35 NeuroMab, California, USA), CASPR2 (1:500; kindly provided by E. Peles), Catalase (1:200; C0979 Sigma-Aldrich, Missouri, USA), CD3 (1:150; MCA1477 Serotec, MorphoSys, Germany), EEA1 (1:300; ab2900 Abcam, Cambridge, United Kingdom), K$_v$1.1 (1:50; sc11184 Santa Cruz Biotechnology, Texas, USA), K$_v$1.1 (1:50; clone 20/78 NeuroMab), K$_v$7.2 (1:2000; PA1-929 ThermoSicentific), LAMP1 (WB, 1:400; IF, 1:200; IEM, 1:200; 553792 BD Bioscience, New Jersey, USA), LIMP-2 (WB, 1:250; IF, 1:2000; kindly provided by J. Blanz), MAC-3 (1:400; 553322 BD Bioscience), Na$_v$1.6 (1:500; ASC-009 Alomone labs, Israel), NF155 (1:1000; kindly provided by P. Brophy), P0 (1:1000 [*Archelos et al., 1993*]); P2 (1:500; sc-49304 Santa Cruz), PLP (1:5000 [*Jung et al., 1996*]), PMP22 (1:1000; SAB4502217 Sigma), PMP70 (1:600; ab3421 Abcam), RAB7 (1:300; R4779 Sigma), TAG-1 (1:500; kindly provided by E. Peles), ßiv spectrin (1:400; kindly provided by M. Rasband), α-Tubulin (1:1000; T 5168 Sigma), III *β*-Tubulin (1:1000; Covance, New Jersey, USA).

## Lysosomal enzyme activity assay

Sciatic nerves were homogenized using Precellys 24 (twice for 20 s at 5000 rpm) in 150 µl ice-cold lysis buffer containing 10 mM Tris-HCl pH 7.4, 150 mM NaCl, 5 mM EDTA, 0.5% Triton X-100, 1 x PEFA, and Protease inhibitors (Complete, Roche). Nerve lysates were centrifuged for 10 min at 15,000 g to remove cellular debris and nuclei. The supernatant was incubated with 10 mM of the substrate (i.e. p-nitrophenyl-a-D-mannopyranosid or p-Nitrophenyl-N-acetyl-ß-D-glucosaminide; Sigma) dissolved in 0.2 M citrate buffer, and the reaction was stopped by addition of 500 µl 0.4 M glycine NaOH (pH 10.4; adjusted with 1 M NaOH) to stop enzyme activity. Samples were centrifuged for 10 min at 15,000 g, and the supernatants of two technical replicates were measured at 405 nm with an Eon microplate spectrophotometer (BioTek, Vermont, USA).

## Quantitative real-time PCR

Total RNA was extracted from sciatic nerve lysates as described previously (*Fünfschilling et al., 2012*). Reactions were performed with four technical replicates of nerves obtained from six different animals per group.

PCR primers were specific for *β-actin* (forward 5′-CTTCCTCCCTGGAGAAGAGC and reverse 5′-ATGCCACAGGATTCCATACC), *Lamp1* (forward 5′-CCTACGAGACTGCGAATGGT and reverse 5′-CCACAAGAACTGCCATTTTC), *Limp-2* (forward 5′-TGGAGATCCTAACGTTGACTTG and reverse 5′-GGCCAGATCCACGACAGT), and *Pex5* (forward 5′-CACATCCGCTTCCTATGACA and reverse 5′-AAAAGGCTGAGGGTGGTCA).

## Mass spectrometry

To quantify PE (diacyl), PE O- (alkanyl-acyl), and PE P- (alkenyl-acyl) species, acidic (PE and PE O-) or neutral (PE P-) lipid extractions were performed as described (*Bligh and Dyer, 1959*). Typically, 1–5 µl of a 1:10 dilution of total lysates were measured. As lipid standards, 50 pmol of PE standards and 40–60 pmol of a PE P- standards were used. Standard syntheses were performed as described (*Özbalci et al., 2013*; *Paltauf and Hermetter, 1994*). After solvent evaporation under a gentle nitrogen flow at 37°C, lipids were re-suspended in 50 µl 10 mM ammonium acetate in methanol. PE/PE O- and PE P- species were analyzed in neutral loss or precursor ion mode selecting for class-specific fragments on a QTRAP5500 (Sciex, Massachusetts, USA) equipped with a NanoMate devise (Advion, New York, USA), employing MS settings as described (*Özbalci et al., 2013*). Data evaluation was done using LipidView (AB Sciex) and an in-house-developed software (ShinyLipids). Species annotated as PE O- mainly contain plasmenylethanoamines but also minor amounts of PE P- species and PE species with odd-numbered fatty acids. Extraction of gangliosides was performed as described (*Sampaio et al., 2011*), with the exception that the first neutral extraction was performed using chloroform:methanol as a 17:1 (v/v) solution, followed by a chloroform:methanol 2:1 (v/v) extraction. For quantification of gangliosides, GD1a and GD1b 50 pmol N-CD3-Stearoyl GM3 and N-CD3-Stearoyl-GM1 (Matreya, Pennsylvania, USA) were used as internal ganglioside standards. Following evaporation, samples were re-suspended in 100 µl methanol. Gangliosides were subjected to UHPLC-MS

analysis, using a CSH C18 column (1 × 150 mm, 1.7 µm particles, Waters) coupled to a QExactive high-resolution Orbitrap mass spectrometer (Thermo) equipped with an ESI source.

For GD1a/b quantification, 30 µl aliquots of the aqueous and the chloroform:methanol (2:1) phase were transferred to Eppendorf cups, evaporated and resuspended in 50 µl buffer containing 60% of mobile phase A (acetonitrile:water; 60:40 (v/v) with 10 mM ammonium formate and 0.1% formic acid) and 40% mobile phase B (isopropanol:acetonitrile, 90:10 (v/v) with 10 mM ammonium formate and 0.1% formic acid). Of each sample, 10 µl was subjected to UPLC separation (Dionex, California, USA), using a multistep gradient. Full MS scans were acquired for 30 min in negative ion mode (600–1800 m/z) with automatic gain control target set to $1 \times 10^6$ ions. The maximum injection time was set to 200 ms and a FHWM resolution of 140,000 (at m/z 200). In addition to Full MS scans, all ion fragmentation scans in negative ion mode were performed at a resolution of 70,000, scanning a mass range of 120–600 m/z with a normalized collision energy set to 30 eV.

Data evaluation of Full-MS scans (profile spectra) was performed using MassMap (MassMap, Germany). Therefore, data files obtained were converted to mzXML-files using the software Proteowizard. mzXML-Files were then converted to mmp-Files using the LC-MS data evaluation software MassMap.

VLCFA determination was performed using 10 µl of a 1:10 dilution of total membrane fractions. 500 pmol of C25 fatty acid (Dr. Ehrenstorfer GmbH, Germany) and C23 fatty acid (Matreya) was used as internal standards. Prior to extraction, glass tubes where washed with chloroform containing 1% acetic acid. The samples were resuspended in 80 µl of toluene. 1 ml acetonitrile-hydrochloric acid (conc.) (4:1) was added and incubated for 2 hr at 90°C. After cooling to room temperature, 2 ml hexane were added, the samples were vortexed, incubated for 5 min and centrifuged for 2 min at 2000 rpm. The upper phase was transferred to a new glass tube. After evaporation of the organic phase under a gentle stream of nitrogen, lipid extracts were resuspended in 100 µl chloroform:methanol:water (50:45:5, v/v/v) with 0.1% ammonium hydroxide solution. Of each sample, 5 µl was transferred to a 96-well plate containing 10 µl 0.01% piperidine in methanol and 5 µl MeOH. Samples were subjected to mass spectrometric analysis on a QSTAR Elite (Sciex) instrument equipped with a NanoMate (Advion). TOF scans were performed over a mass range of 100–500 m/z in negative ion mode. Data sets were processed and evaluated using LipidView (Sciex).

## Electrophysiology, ex vivo

Following cervical dislocation mouse sciatic nerves were rapidly transferred into a perfusion chamber filled with gassed ACSF at 37°C containing in mM: 126 NaCl, 3 KCl, 26 NaHCO$_3$, 1.25 NaH2PO$_4$, 25 glucose, 2.5 MgSO$_4$ and 2 CaCl$_2$; a pH of 7.4 was ensured by continuous gassing with carbogen (95% O2/5% CO2). Nerves' lengths were measured and were then allowed to adapt for approximately 30 min before electrophysiological recordings began. Suction electrodes backfilled with ACSF were used for stimulation and recording. Electrical stimulation was performed at different intensities ranging between 0.13 mA and 0.3 mA, and increased manually in step-width of 0.05 mA. Inter-stimulus intervals were 3 s and each of the stimulus intensities was used 10 times. CAPs were continuously measured. Recorded signals were acquired at 100 kHz, amplified x100 by Ext-2F (NPI Electronic, Germany), and a further 50-fold by SR560 (Stanford Research Systems, California, USA), and filtered. Recordings were controlled by patchmaster software (HEKA, Germany) with an EPC9 amplifier interface (HEKA). Stimulus intensities were altered using Stimulus Isolator A385 (World Precision Instruments).

Analysis of electrophysiology was performed using Matlab. Each nerve was analyzed separately. Responses triggered but the same stimulus repetitions were averaged together. Nerve conduction velocity was calculated for each of the stimulus intensities as the mean across trials of the ratio between the length of the nerve and the time between peaks of stimulus-artifact (first small positive peak) and response (third positive peak).

## Electrophysiology, in vivo

Mice aged 9 months were anesthetized with ketaminhydrochloride (100 mg kg$^{-1}$)/xylazin hydrochloride (8 mg kg$^{-1}$). Steel needle electrodes were placed subcutaneously, one pair at sciatic notch (proximal stimulation), a second pair at the tibial nerve above the ankle (distal stimulation).

Supramaximal square wave pulses lasting 100 ms were delivered using a Toennies Neuroscreen (Jaeger, Germany). Compound muscle action potential (CMAP) was recorded from the intrinsic foot muscles.

## Image processing

Digital images were processed and quantified with ZEN 2012 software (Zeiss) and/or Fiji.

## Statistical analysis

All numerical values are shown as the mean ± s.e.m.; n = 3–6, unless stated otherwise. Statistical significance was determined using GraphPad Prism5 by two-way ANOVA or the two-tailed Student's t test for unpaired samples assuming unequal variance and p values below 0.05 were marked as significant: *$p < 0.05$; **$p < 0.01$; ***$p < 0.001$). Normal distribution was assumed, but not formally tested.

## Study approval

Mice were kept in the mouse keeping facility of the Max-Planck-Institute of Experimental Medicine at 12 hr light/dark cycle. All experiments were executed according to the German Animal Protection Law and approved by the government agency of the State of Lower Saxony, Germany.

## Acknowledgements

We thank E Peles, M Rasband, and P Brophy for providing antibodies, Steve Scherer for helpful comments on the manuscript, and we acknowledge P Saftig, R A Nixon and members of the Department of Neurogenetics for helpful discussions. We thank J Günther, T Ruhwedel, and A Fahrenholz for excellent technical support. BB is supported by the DFG (TRR83) and is investigator of the excellence cluster CellNetworks (EXC81). CMK was supported by grants from Oliver's Army (GB) and The Myelin Project (USA). KAN is supported by the DFG (Research Center Nanoscale Microscopy and Molecular Physiology of the Brain and TR43) and holds an ERC Advanced Grant (AxoGLIA, MyeliNANO).

## Additional information

### Competing interests

K-AN: Reviewing editor, *eLife*. The other authors declare that no competing interests exist.

### Funding

| Funder | Grant reference number | Author |
|---|---|---|
| Deutsche Forschungsgemeinschaft | TR43 | Klaus-Armin Nave |
| European Commission | ERC advanced grant - AxoGLIA | Klaus-Armin Nave |
| European Commission | ERC advanced grant - MyeliNANO | Klaus-Armin Nave |
| Olivers Army | Personal grant | Celia Michèle Kassmann |
| Myelin Project | Augusto Odone Young Investigator Award | Celia Michèle Kassmann |
| Deutsche Forschungsgemeinschaft | TRR83 | Britta Brügger |

The funders had no role in study design, data collection and interpretation, or the decision to submit the work for publication.

### Author contributions

SK, Mouse breeding, histology, light microscopy, data analysis, incl. statistics, and interpretation of data; SR, Conceptualization, Investigation, Mouse breeding, histology, light microscopy, data analysis, incl. statistics, and interpretation of data; LdH, Formal analysis, Methodology, Data analysis, incl.

statistics, and interpretation of electrophysiology data; BB, Software, Formal analysis, Funding acquisition, Investigation, Methodology, Lipid mass spectrometry: data analysis, incl. statistics, and interpretationof data; TK, Conceptualization, Investigation, Histology, light microscopy, data analysis, incl. statistics, and interpretation of data; EA, Software, Investigation, Methodology; SQ, Formal analysis, Investigation, Methodology; JB, HW, Formal analysis, Methodology; RM, Formal analysis, Validation, Investigation, Methodology; KN, Investigation, Visualization, Methodology; MWS, Formal analysis, Methodology, Data analysis, incl. statistics, and interpretation ofelectrophysiology data; TS, CL, WM, Investigation, Methodology; MB, Resources, Provided floxed mice andrevised the article critically for important intellectual content; K-AN, Funding acquisition, Writing—review and editing, Revising thearticle critically for important intellectual content, Final approval of the version to be published; CMK, C.M.K. supervised theproject, designed experiments, analyzed data, and wrote the manuscript

### Author ORCIDs
Wiebke Möbius, http://orcid.org/0000-0002-2902-7165
Klaus-Armin Nave, http://orcid.org/0000-0001-8724-9666
Celia Michèle Kassmann, http://orcid.org/0000-0003-0993-9455

### Ethics
Animal experimentation: Mice were kept in the mouse keeping facility of the Max Planck Institute of Experimental Medicine at 12 hr light/dark cycle. All experiments were executed according to the German Animal Protection Law and approved (permit numbers 33.9-42502-04-10/0313; 33.19-42502-04-15/2006) by the government agency of the State of Lower Saxony, Germany.

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
