## [Decision Letter]

Thank you for submitting your article "Peroxisomal dysfunctions cause storage of lysosomes and axonal Kv1 channel redistribution in peripheral neuropathy" for consideration by *eLife*. Your article has been favorably evaluated by Gary Westbrook (Senior Editor) and three reviewers, one of whom is a member of our Board of Reviewing Editors. The reviewers have opted to remain anonymous. The reviewers have discussed the reviews with one another and the Reviewing Editor has drafted this decision to help you prepare a revised submission.

Summary:

In this short report, Kleinecke et al. describe the consequences of peroxisome dysfunction upon *Pex5* deletion in Schwann cells. The phenotype is strikingly less severe than that in that observed when *Pex5* is deleted in oligodendrocytes as described in a previous paper from this group, and interestingly suggests peroxisome dysfunction has severe consequences on the function of neighboring lysosomes. Indeed, the authors demonstrate that there is mislocalization of lysosomal proteins, gangliosides and other proteins including K_v_1.1 channels. The authors suggest that these changes have important effects on nerve physiology. Lastly, some of the observed phenotypes were also observed in mice deficient in *Abcd1*. The authors propose an interesting model that suggests lipids on Schwann cell membranes stabilize certain adhesion proteins and their associated ion channels. Disruption of these lipids leads to K_v_1.1 channel mislocalization which in turn is proposed to lead to nerve conduction changes.

Essential revisions:

Although the presented data suggest very interesting models, the correlative nature of this data could be strengthened by additional experiments.

1) The examination of conditional knockouts for the peroxisomal protein *Mfp2* for similar phenotypes that were observed in *Pex5* cKOs would strengthen the conclusions and provide some mechanistic insight. For example, if many of the findings observed in the *Pex5* cKOs, particularly the abnormal CAP recordings, could also be demonstrated in the *Mfp2* cKOs, it could be said with greater confidence that manipulation of lipid metabolism is specifically producing these changes. Similarly, the connection to human disease could be strengthened if the CAP recording abnormalities are also found in *Abcd1* KOs.

2) Additional studies on nerve physiology are needed. The studies on nerve conduction show clear changes in conduction velocity and CAP amplitude. The authors conclude this is a function of ectopic internodal K^+^ channels leading to conduction block. Some additional experiments are needed to support these conclusions. Decreased internodal distance could also decrease conduction velocity, whether this could be a progressive change is unclear, but it should be ruled out. Conduction block is typically defined clinically by stimulating at a distal point and a proximal point, and comparing amplitudes. If the distal amplitude is greater than the proximal, this is evidence of conduction block (increased failure to propagate and action potential with increased distance). This should be tested. It is unclear that the decrease in CAP could not be simply the result of lower amplitude action potentials, or a complete failure in excitability of a subset of axons (larger, faster conducting axons, for example, which could also explain the velocity changes).

3) Furthermore, examination of the localization of K_v_1.1 channels should be done at an earlier time point. It's not clear from the data whether membrane complexes break off and diffuse into the adjacent internodal and paranodal regions, or if localization is simply misspecified during the initial formation of myelination. Although the progressive mislocalization of K_v_1.1 channels to the internode with age supports the authors' model, it would be helpful to examine an earlier time point when sciatic nerve myelination has just completed.

4) Lastly, to strengthen the proposal that ganglioside accumulation underlies mislocalization of K_v_1.1, quantitation of the GPI-anchored protein TAG-1 and CASPR2 would be useful to rule out for possible alterations in expression that may be altered by lipid metabolism.

---

## [Author Response]

*Essential revisions:*

*Although the presented data suggest very interesting models, the correlative nature of this data could be strengthened by additional experiments.*

*1) The examination of conditional knockouts for the peroxisomal protein Mfp2 for similar phenotypes that were observed in Pex5 cKOs would strengthen the conclusions and provide some mechanistic insight. For example, if many of the findings observed in the Pex5 cKOs, particularly the abnormal CAP recordings, could also be demonstrated in the Mfp2 cKOs, it could be said with greater confidence that manipulation of lipid metabolism is specifically producing these changes. Similarly, the connection to human disease could be strengthened if the CAP recording abnormalities are also found in Abcd1 KOs.*

We agree that a similar pathology of PEX5 and MFP2 cKO nerves would strengthen our conclusions. We have therefore analyzed sciatic nerves from aged (16 months old) MFP2 mutants, which demonstrated nearly 3-fold lysosomal enzyme activity (new Figure 4—figure supplement 1). Moreover, electrophysiological recordings revealed reduced CAPs of mutants compared to age-matched controls, as predicted. (new Figure 4—figure supplement 1).

We also performed the requested electrophysiological recordings of ABCD1 animals. At age 2 months, there were no differences yet compared to controls. This was not unexpected, because the mislocalization of K_v_1 potassium channels at that age affected only a minor portion of internodes (original Figure 4). We note, that in human AMN patients the corresponding PNS symptoms will not manifest before the 2^nd^ to 3^rd^ decade of life, and it is well known that mouse and man differ in disease expression and disease kinetics (specifically for mutations of the X-linked ABCD1 gene; Ferrer et al., Brain Pathol., 2010). Indeed, ABCD1 knockout mice showed a significant mislocalization of potassium channels only at 22 months of age (original Figure 4), which corresponds to the human 7^th^ – 8^th^ decade of life. At that age we measured reduced compound muscle action potential (CMAP) compared to younger mutants, however there was already a major overlap with other aging-dependent changes of conductivity that also affect wild-type mice. Thus, it was not possible to simply separate nerve aging effects from ABCD1 genotype-dependent changes of conductivity. Although, we noted a clear trend (P=0.12) between the groups, we prefer to not add these data into the current manuscript. Waiting for another cohort of ABCD1 mutant with the age of 22 months is unfortunately not possible within the time frame of the revision.

*2) Additional studies on nerve physiology are needed. The studies on nerve conduction show clear changes in conduction velocity and CAP amplitude. The authors conclude this is a function of ectopic internodal K^+^ channels leading to conduction block. Some additional experiments are needed to support these conclusions. Decreased internodal distance could also decrease conduction velocity, whether this could be a progressive change is unclear, but it should be ruled out. Conduction block is typically defined clinically by stimulating at a distal point and a proximal point, and comparing amplitudes. If the distal amplitude is greater than the proximal, this is evidence of conduction block (increased failure to propagate and action potential with increased distance). This should be tested. It is unclear that the decrease in CAP could not be simply the result of lower amplitude action potentials, or a complete failure in excitability of a subset of axons (larger, faster conducting axons, for example, which could also explain the velocity changes).*

As requested, we have addressed the possibility that conduction blocks are responsible for the reduced CAPs in PEX5 mutant nerves. Here, we have turned to in vivo analysis and have assessed the ratio of compound muscle action potentials (CMAP) after proximal and distal nerve stimulation. As shown in new Figure 1—figure supplement 2, the distal amplitude was significantly (p<0.01) greater than the proximal amplitude, as predicted for axonal conduction blocks. A sentence referring to these new data has been added in the third paragraph of the Results.

To investigate a possible impact of altered internodal length on conduction velocity (NCV) we analyzed PEX5 mutant nerves at age 2 months. Here, the average internodal length of mutant fibers (623 µm) was approximately 10% shorter compared to controls (691 µm), as summarized in the fourth paragraph of the Results and depicted in new Figure 1—figure supplement 3. Although significant, this length difference should by itself not result in slowed conduction, because shortening the internodal length accelerates nerve conduction to a flat maximum, as shown by the Brophy lab. That paper demonstrated unaltered NCV despite a 30% reduced internodal length in the ΔPDZ-Prx mouse mutant (Wu et al., Curr. Biol., 2012).

*3) Furthermore, examination of the localization of K_v_1.1 channels should be done at an earlier time point. It's not clear from the data whether membrane complexes break off and diffuse into the adjacent internodal and paranodal regions, or if localization is simply misspecified during the initial formation of myelination. Although the progressive mislocalization of K_v_1.1 channels to the internode with age supports the authors' model, it would be helpful to examine an earlier time point when sciatic nerve myelination has just completed.*

For the original submission we had analyzed K_v_1.1 in PEX5 mutant and control nerves at P19 (original Figure 2), i.e. shortly after juxtaparanodes have formed in development. As requested, we have extended the observational time points to P17 and then to P7.

At P7 and also at P17 K_v_1.1 appeared not yet completely clustered, and in the majority of myelinated fibers (mutants and controls) the staining included a diffuse signal in nodal/paranodal and paranodal/ juxtaparanodal regions. In these immature fibers that lacked regular K_v_1.1-positive juxtaparanodes, we never observed abnormal internodal clusters.

Likewise, in ABCD1-mutant nerves a comparison of young (2mo) and old (22 mo) sciatic nerves revealed significant internodal clusters only at age 22 months (as shown in Figure 4). This strengthens our model of an initially normal formation of juxtaparanodes and a secondary breaking off. We have added a clarifying sentence in the last paragraph of the Results.

*4) Lastly, to strengthen the proposal that ganglioside accumulation underlies mislocalization of K_v_1.1, quantitation of the GP1-anchored protein TAG-1 and CASPR2 would be useful to rule out for possible alterations in expression that may be altered by lipid metabolism.*

We agree that alterations of a cell's lipid metabolism could affect (e.g. SREBP-dependent) gene expression and result in overexpression as a hypothetical cause of ectopic protein localization. We have therefore analyzed the mRNA expression levels of TAG-1, which is the K_v_1.1 anchor expressed by Schwann cells, but could not detect significant differences (p=0.42) as shown in NEW Figure 2. Note that CASPR2 is expressed in the neuronal/axonal compartment.